# Endogenous Retroviral Elements in Human Development and Central Nervous System Embryonal Tumors

**DOI:** 10.3390/jpm11121332

**Published:** 2021-12-08

**Authors:** Tara T. Doucet-O’Hare, Jared S. Rosenblum, Ashish H. Shah, Mark R. Gilbert, Zhengping Zhuang

**Affiliations:** 1National Cancer Institute, Center for Cancer Research, Neuro-Oncology Branch, Building 37, Room 1000, 37 Convent Drive, Bethesda, MD 20892, USA; doucettt@nih.gov (T.T.D.-O.); jared.rosenblum@nih.gov (J.S.R.); mark.gilbert@nih.gov (M.R.G.); 2National Institutes of Neurological Disorders and Stroke (NINDS), Surgical Neurology Branch, Bethesda, MD 20892, USA; ashish.shah2@nih.gov

**Keywords:** endogenous retrovirus, HERV-K, HML-2, developmental tumors, AT/RT, brain, tumors

## Abstract

Human endogenous retroviruses (HERVs), which are critical to normal embryologic development and downregulated during normal maturation, have been implicated in a variety of cancers. Abnormal persistent production of HERVs has been suggested to play a role in oncogenesis and to confer stem cell properties to cells. We recently demonstrated that the most recently incorporated HERV element (HERV-K HML-2) has been associated with the pathogenesis of the embryonal atypical teratoid rhabdoid tumor (AT/RT), shifting our understanding of embryonal tumor development. HML-2 expression is vital for proper human development and its expression is suppressed via methylation or chromatin remodeling as cells differentiate. We previously found that dysfunctional chromatin remodeling due to loss of SMARCB1 expression induces HML-2 envelope (env) expression, impairing cellular differentiation and migration, and facilitating tumor growth in AT/RT. Epigenetic dysregulation in other embryonal tumors with concomitant expression of stem-cell markers may facilitate HML-2 expression. Future studies could utilize HML-2 as potential diagnostic criteria, use its expression as a treatment biomarker, and investigate the efficacy of therapies targeting cells with high HML-2 expression.

## 1. Discovery of Endogenous Retroviruses

Endogenous retroviruses have been incorporated into the genomes of many organisms throughout evolution by inserting into the genomes of germ cells and thereafter being inherited in the same way as Mendelian genes [1]. Retroviruses were only named in 1974 [2]; however, their existence was detected in the early nineteenth century by cattle and sheep farmers who observed a pathogenic effect that unidentified retroviruses had on their livestock [3]. In 1908, erythroleukemia, a subtype of acute myeloid leukemia, was shown to be experimentally transmissible between chickens, establishing the first connection between oncogenesis and retroviruses [3]. A discovery by Rous in 1961 revealed that these viral particles contained RNA; however, the cells transformed by the virus retained stable properties through many cell divisions [4]. A ‘virus-transformed phenotype’ appeared to be heritable in the absence of viral replication [5]. This observation later led Temin to hypothesize that the RNA Rous sarcoma virus (RSV) made a DNA copy that could integrate into the host genome [6]. Although it was accepted that DNA tumor viruses could integrate into the genomes of somatic cells, many regarded the idea of transmission through the germline of healthy animals as unlikely. In 1966, a retroviral protein was detected by a serological assay developed for poultry farming in chickens without active infections of avian leukosis virus (ALV) [7]. Finally, in 1970, two publications in Nature revealed the existence of the reverse transcriptase enzyme, capable of reverse transcribing DNA from an RNA precursor [8,9]. Taken together, this work revealed endogenous retroviruses integrated into animal genomes and further demonstrated their Mendelian inheritance pattern due to their incorporation into germ cells.

## 2. Expression/Regulation of HML-2 in Early Development

Although the validity of endogenous retroviruses was no longer in question, it was still uncertain what, if any, role they played in the life cycle of an organism aside from pathogenesis. It was only after the initial sequencing of the human genome was completed that we began to appreciate a large percent of our genome originated from the activity of endogenous retroviruses [10]. Reverse transcriptase is responsible for more than 40% of most mammalian genomes and its products have contributed to the genomes of many organisms, from yeast to mammals [11,12]. Today we know that several endogenous retroviruses have become adapted to serve various functions in mammalian development. For example, one co-opted endogenous retrovirus, the human endogenous retrovirus W produces an envelope protein, syncytin-1, which is vital for placental formation and normal embryonal and fetal development [13,14].

Human endogenous retrovirus K (HERV-K), subtype HML-2, incorporated into the germline of ancestors of Homo sapiens millions of years ago and have been maintained in our genome [15]. These elements have been noted to be highly expressed in pluripotent stem cells and in germ-cell tumors likely due to their undifferentiated nature and embryonal origin [16,17,18]. There are approximately 90 intact HML-2 proviruses which encode one or more viral proteins including a pathogenic envelope (env) protein [19,20]. The promoter of HML-2 is a long terminal repeat (LTR) which sits 5′ or upstream of the proviral sequence [21,22,23,24]. HML-2 gene regulation is controlled by several mechanisms, including methylation of its promoter (5′ LTR), chromatin remodeling, binding by transcription factors, and repression via transfer RNA derived fragments (tRFs) [21,22,23,24,25].

HML-2 expression is vital for proper human development and its expression is normally highly controlled throughout the process [14]. HML-2 expression starts at the eight-cell stage of embryo development post-fertilization, continues in preimplantation blastocysts, and decreases following stem cell derivation from blastocyst outgrowths [14]. Consistent with previous findings of physiologic HERV expression (HERV-W and HML-2) in undifferentiated cellular states, the highest HML-2 expression occurs in the epiblast [13,14].

In a sequencing study of DNase hypersensitivity sites, OCT4 binding motifs were noted to be enriched during zygotic genome activation in humans but not in mice [26]. The presence of OCT4 binding motifs in DNAse sensitive areas indicates that, during human zygote development, pluripotency gene promoters are available for transcription factor binding and activate transcription. HML-2 has multiple binding sites for transcriptional activator OCT4, and its expression is regulated by OCT4 binding during early development [14]. This suggests that HML-2 transcription coincides with other pluripotent genes during development and in undifferentiated cells.

## 3. HERV-K Expression Is Associated with Stemness Markers

We recently found that HML-2 loci are highly expressed in pluripotent stem cells and their expression gradually decreases as they differentiate [22]. We also elucidated that the expression of the HML-2 env was critical for cells to maintain stemness via an interaction at the cell membrane [22]. Interactions between HML-2 env and its membrane receptor, CD98HC, stimulate downstream mammalian target of rapamycin (mTOR) signaling pathways and lysophosphatidylcholine acyltransferase (LPCAT1) mediated epigenetic changes which allows cells to retain their stemness [22].

Downregulation of HML-2 expression through chromatin remodeling or methylation of its promoter, 5′ LTR, occurs as cells differentiate into neurons (Figure 1) [22]. When HML-2 expression occurs in differentiated cortical neurons, the env protein causes cytotoxicity and has been linked with creating neurodegenerative phenotypes in mice [27]. HML-2 transcripts and proteins have also been detected in brains from patients with amyotrophic lateral sclerosis (ALS), a progressive fatal neurodegenerative disorder [27]. Recently, HML-2 transcription has been associated with *NTRK3* (neurotrophic receptor tyrosine kinase) hyperactivity and resulting in impaired cortical neuronal differentiation; reduction of *NTRK3* expression restored normal differentiation [28]. *NTRK3* is a kinase activated by neurotrophin binding that is integral in the signaling cascade for neuronal differentiation [29]. Thus, the regulation of HML-2 expression must be carefully regulated for cells to either retain stemness or differentiate and remain viable after differentiation.

HML-2 env expression occurs in many tumor types and has been shown to imbue cancer cells with stem-cell-like characteristics [22,30,31]. We recently found high HML-2 env expression confers a stem-cell-like phenotype and contributes to active proliferation in atypical teratoid rhabdoid tumor (AT/RT) [30]. We observed co-expression of *OCT4* and HML-2 env in AT/RT cells, further cementing the connection between the stem-cell transcription factor and expression of HML-2 transcripts [30]. Similarly, we found that there is robust concomitant expression of the neural stem cell marker Nestin in cells expressing high HML-2 env protein in AT/RT patient-derived cell lines [30]. We also found that targeting HML-2 env expression in AT/RT resulted in significantly lower cell proliferation and disrupted the cell–cell communication required for cells to maintain pluripotency [30].

Targeted HML-2 downregulation in metastatic breast cancer and in pancreatic cancer resulted in a concurrent decrease in NRAS protein suggesting HML-2 may target Ras genes [32,33]. When patient derived AT/RT cell lines were treated with shRNA targeting HML-2 env, a significant decrease in both env and *N-RAS* transcription occurred [30]. When the AT/RT cells were co-transfected with a CRISPRi construct (+gRNA targeting HML-2 LTR) and an *N-RAS* plasmid, HML-2 expression was higher by qRT-PCR compared with CRISPRi (+gRNA) and pcDNA *CAT* co-transfection, which served as a control [30]. Thus, AT/RT cells with *N-RAS* overexpression were able to overcome the negative consequences, lower cell proliferation and decreased viability, of HML-2 downregulation via CRISPRi. Our results showed that *N-RAS* overexpression was sufficient to restore HML-2 transcription; moreover, at five days post-transfection, we observed significantly more viable cells in the cells transfected with *N-RAS* as compared to pcDNA *CAT* [30]. We also observed larger cell aggregates resulting from transfection with *N-RAS*, further supporting the restoration of cell proliferation in the cell lines [30]. From our study with patient-derived AT/RT cell lines, HML-2 expression clearly confers a growth advantage via the RAS regulatory pathway [30].

## 4. HML-2 Expression in Cancer

Expression of endogenous retroviral elements have been associated with disease progression in a variety of cancers including melanoma, ovarian cancer, and hepatocellular carcinoma [23,34,35,36]. HML-2 expression upregulation has been found in tumors with marked epigenetic dysregulation [24,30,37]. In a recent study of repetitive element methylation and expression in colon cancer, the HML-2 promoter (5′ LTR) was hypomethylated in tumor tissue, resulting in higher transcription compared to normal tissue; further, the env protein was only expressed in the tumor [24].

A recent study of rhabdoid tumors detected SMARCB1-dependent (SWI/SNF related, matrix associated, actin dependent regulator of chromatin, subfamily B, member 1) re-expression of endogenous retroviruses (ERVs) along with interferon-signaling [37]. In this study, the authors observed a subset of ERV loci were actively expressed and hypothesized that this expression could lead to an anti-tumor response via viral mimicry [37]. We were the first to demonstrate the mechanism by which HERV-K HML-2 upregulation occurs in a type of rhabdoid tumor, namely AT/RT [30]. We found that 95% of AT/RT tumors express HERV-K (HML-2) env and that, in vitro, the cells depend on its continued expression for survival and proliferation [30]. Our findings are the first observation of HML-2 dysregulation in embryonal tumors of the CNS [30].

Nearly all AT/RT cases are caused by a loss of SMARCB1 expression and are comprised of undifferentiated cells which form tumors due to failure of proper differentiation and migration [38]. A small percentage of AT/RT cases occur due to the loss of SMARCA4, another core member of the SWI/SNF chromatin remodeling family; moreover, single inherited mutations of either SMARCB1 or SMARCA4 contribute to rhabdoid tumor predisposition syndrome (RTPS) 1 or 2, respectively [39]. Although no one has evaluated the expression of endogenous retroviral elements in individuals with RTPS1/2, from our work we hypothesize these individuals are likely to have an upregulation of HML-2 transcription due to the absence of functional SWI/SNF regulation at HML-2 LTRs. In addition, we expect HML-2 upregulation will be a feature of non-CNS tumors which lack SMARCB1 expression, especially in tumors which express C-MYC.

AT/RT cells typically express a mixture of epithelial, mesenchymal, and neuroectodermal genes and appear as small cells with a high nuclear to cytoplasm ratio [38]. We found that in the absence of SMARCB1 expression, aberrant chromatin remodeling occurs leading to subsequent continued expression of pluripotency genes such as OCT4 and HERV-K (HML-2) env [30]. Without functional SMARCB1, the HML-2 long terminal repeat (LTR) is bound by C-MYC, a transcriptional activator, and HML-2 transcripts are produced [30]. When SMARCB1 is expressed, it binds to C-MYC and prevents it from interacting with the HML-2 LTR resulting in decreased transcription (Figure 2) [30]. These findings coupled with others suggest a possible role for HML-2 in tumors which possess an immature undifferentiated morphology such as AT/RT.

Overexpression of SMARCB1 in the patient derived AT/RT tumor cells led to significantly reduced HML-2 transcription [30]. Using chromatin immunoprecipitation (ChIP), we observed significantly higher binding of SMARCB1 to the HML-2 promoter than to the hypoxanthine–guanine phosphoribosyltransferase *(HPRT)* promoter [30]. From these studies, the expression and normal function of SMARCB1 is a clear determinant of HML-2 expression.

Identification of the HML-2 loci in the genome expressed in various tumor types is important for understanding the epigenetic implications of their expression as well as identifying which loci may be involved. In AT/RT patient cells, we identified expression from 36 loci from nearly all chromosomes with the exceptions of 14, 15, 18, and 21 [30]. The most actively transcribed loci were on chromosomes 1 and 19; however, of all expressed loci, only three encoded a full length and possibly functional env protein: transcripts originating from Chr19q11, Chr7pp22.1a, and Chr7p22.1b [30]. When seeking to validate the envelope transcript identified by RNA-sequencing, the reverse transcribed cDNA from a cell line or sample can be cloned into a vector [30]. Using Sanger sequencing, we were able to obtain longer reads than with RNA-Seq analysis (700 vs 100 nucleotides) which enabled better alignment of the expressed transcripts to the genome [30]. This technique should be employed in additional studies of HML-2 expression in cancer to precisely identify loci which can encode functional HML-2 proteins, such as env, that contribute to tumorigenesis.

We have not evaluated the potential contribution of HML-2 transcription from polymorphic elements; however, in large granular lymphocytic (LGL) leukemia, patients have a heavier genomic load of polymorphic HML-2 loci compared to those with a similar genetic background [20]. A study similar to the LGL leukemia patient analysis could reveal the potential importance of polymorphic HML-2 loci in AT/RT and further identify proviruses which produce the env protein. Due to the role of SMARCB1 in the downregulation of HML-2 LTR transcription, any effect on polymorphic HML-2 insertions would not differ from its effect on fixed HML-2 insertions. Therefore, we hypothesize that regardless of the origin of an HML-2 transcript, either from a polymorphic or a fixed provirus, SMARCB1 loss would result in higher transcription of any proviruses with C-MYC transcription factor binding sites. 

In germ-cell tumors originating in the testes, retroviral particles form and bud from the plasma membrane, albeit at a low frequency, after treatment with 5-iodo-2′-deoxyuridine and dexamethasone [40]. HML-2 expression occurs not only in testicular germ-cell tumors, but also in their cellular precursor, carcinoma in situ [41]. In another study, the authors found that the locus that produces the retroviral particles in Tera1 cells was Chr22q11.21 and that HML-2 gag and env transcripts were selectively packaged into the particles [42]. In one study of patients with germ-cell tumors, 85% of patients produced antibodies directed against the env protein, and the authors concluded that anti-transmembrane env antibodies are specific for the tumor [43]. Antibodies against the N terminal of HML-2 gag protein were also detected in 2–4% of patients with seminoma and an 80 kD gag protein was observed in the cytoplasm of seminoma cells [16]. These antibodies may represent biomarkers that could be used for clinical detection, surveillance of progression, and treatment response.

In effect, many aspects of embryonic development regulate the expression of HML-2, and, when any of these canonical pathways are disrupted, it can result in continued and aberrant HML-2 expression. Due to the pervasive expression of HML-2 in early development in stem cells and in undifferentiated tumors, we suggest its expression could be utilized for identifying and targeting tumor cells in certain cancers. Recent biological insights highlighting shared features of embryonal tumors necessitate investigation into mechanisms of persistence of embryonic tissue that becomes cancer. It is critical to find ways to target the cancer cells while doing as little damage to normal tissue as possible, especially in young individuals. Further, certain treatments used for these tumors, such as radiation, can lead to additional mutations and tumor recurrence later in life. Conventional treatments for CNS embryonal tumors including systemic chemotherapy and radiation may confer significant toxicities including radiation-induced neurotoxicity, leukopenia, and development of radiation-induced de novo tumors [44,45]. Therefore, targeted treatments for CNS embryonal tumors are critical in improving outcomes and reducing treatment-related toxicities. We suggest that HML-2 expression may have diagnostic potential, could serve as a marker for treatment efficacy, and may be an excellent novel therapeutic target in CNS embryonal tumors, as is the case for AT/RT.

## 5. Endogenous Retroviral Element Expression as a Target for Treatment

Since many undifferentiated cancers possess HERV antigens, HERV expression has been suggested as a potential tumor-specific therapeutic target [46,47]. In one study, in tumors with high HML-2 expression, env directed antibody treatment reduced metastasis and tumor growth, while inducing an immunogenic response [48]. Additionally, Feng-Johanning et al. demonstrated that HML-2 specific T cells may induce robust anti-tumor immune responses in ovarian and breast cancers [49]. Similarly, a clinical trial utilizing chimeric antigen receptor T cells (CAR-T) against a novel HERV epitope (HERV-E) has also been initiated, with strong preclinical data suggesting that HERVs may serve as immunogenic antigens for certain cancers [50,51].

Aside from therapies against HERV antigens, therapies against the reverse transcriptase enzyme have also been suggested [36,52]. Several studies have demonstrated that high expression of reverse transcriptase occurs in early malignant tissue samples in immunocompetent hosts, which could be regarded as a surrogate for overall HERV expression [36,51]. Therefore, reverse transcriptase inhibitors have been suggested as a potential adjuvant treatment option for certain cancers to target aberrant HERV expression.

In medulloblastoma, reverse transcriptase inhibition promoted cell senescence and induced apoptosis and cellular differentiation [53]. This suggests that reverse transcriptase inhibition may be a personalized therapeutic option for tumors with high reverse transcriptase enzyme production. In a recent study using anti-retroviral drugs, the authors found increased responsiveness of cancer cells with stemness features, which suggests these drugs may represent a new approach to treating aggressive tumors in combination with other chemotherapeutic or radiotherapy regimens [54].

In our study of AT/RT, we observed that dCas9 coupled with suppressor proteins and guide RNA targeting HML-2 sequences was a successful strategy for not only downregulation of HML-2 transcription, but also for decreasing cell proliferation and viability [30]. In another study using SaCas9 which targeted the env gene, the disruption of the env gene in vitro led to decreased transcription and translation of env protein [55]. The disruption of env expression also resulted in changes to regulation of cell proliferation and expression of additional genes involved in RNA binding and alternative splicing such as epidermal growth factor receptor (EGF-R) and nuclear factor kappa-light-chain-enhancer of activated B cells (NF-kB) [55]. Disruption of env expression in AT/RT and prostate cancer cells affects multiple pathways indicating that expression of HML-2 is necessary for their cell viability and proliferation.

## 6. Endogenous Retroviral Element Expression as a Clinical Prognosticator

Several studies have demonstrated that endogenous retroviruses could serve as a biomarker of treatment response (Table 1) [23,42,56]. In melanoma and ovarian cancer, HML-2 hypomethylation was an independent predictor of disease progression and resistance to conventional chemotherapy [23]. For germ-cell tumors, the presence of HML-2 specific gag/env antibodies was also indicative of chemotherapeutic responses [56]. In another study, patients with germ-cell tumors produced fewer antibodies against HML-2 env after anti-tumor treatment [42]. Another study of germ-cell tumors found that the level of HML-2 antibody was stable or elevated post-treatment in 42 patients where progression or relapse occurred [56]. The continued production of antibodies against HML-2 proteins could suggest that, despite treatment, viable tumor cells in residual masses or metastatic foci remain.

Tumor Cancer Genome Atlas (TCGA) studies have suggested that high expression of HML-2 elements was associated with poor survival in several cancers, independent of conventional histopathological grading systems [23,34]. When evaluating the expression of repetitive elements in RNA sequencing datasets, it is important that samples be sequenced at a sufficient depth and undergo ribosomal RNA depletion prior to sequencing to ensure analysis accuracy. Although, the data included in TCGA are limited in this regard, several studies have elucidated interesting observations about retroviral element expression in cancer. In hepatocellular carcinoma, the level of HML-2 transcription was associated with tumor differentiation and patients with higher expression had a poorer prognosis [56]. In another study, HML-2 transcription was upregulated from multiple loci in pediatric hepatoblastoma tumors compared to control tissue from healthy livers [57]; in melanoma, hypomethylation of HML-2 was associated with patients who had reduced disease-free survival [23].

## 7. Conclusions

Human endogenous retroviruses have been a part of the human genome for millions of years and are now integral to our reproduction. However, when the expression of HERVs such as HML-2 are disrupted, they can lead to pathology. Embryonal tumors of the CNS begin during early development and have few targeted treatments. Continued research that identifies the molecular differences between the undifferentiated tumors of the CNS will provide a better understanding of these tumors and inform future diagnostic practices and treatment options. We observed the first evidence of HML-2 expression in the embryonal tumor, AT/RT, and we suggest that its expression may be (1) a useful diagnostic tool and biomarker for surveillance of progression or treatment response and (2) an excellent new therapeutic target.

## Figures and Tables

**Figure 1 jpm-11-01332-f001:**
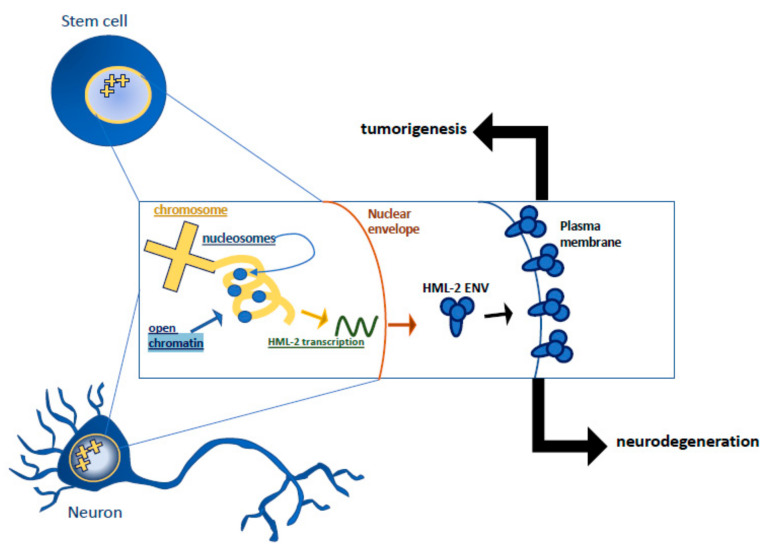
Open chromatin leads to HML-2 expression. HML-2 env protein is translated and translocated to the plasma membrane conferring stem-cell-like phenotypes or can result in neurotoxicity and lead to neurodegeneration.

**Figure 2 jpm-11-01332-f002:**
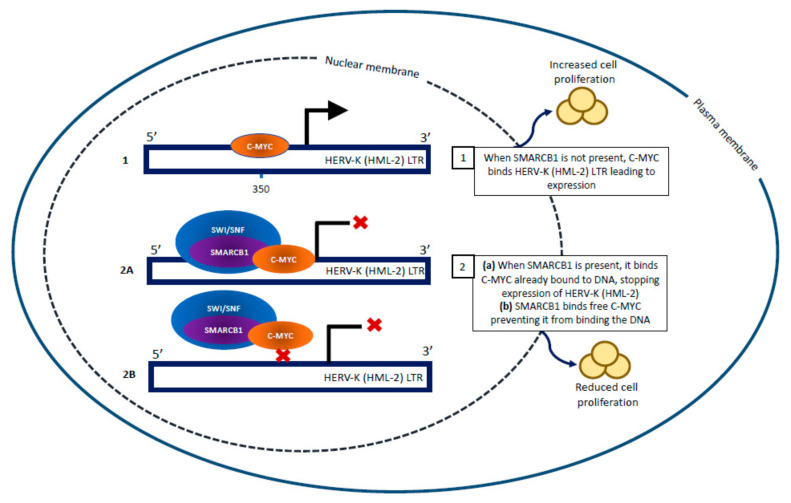
Mechanism of SMARCB1 control of HML-2 transcription. In the absence if the SMARCB1 protein, HML-2 is actively expressed when C-MYC is bound to the LTR (promoter) (1). When SMARCB1 is present, it inhibits C-MYC activation of HML-2 transcription (2 A, B). Figure adapted from previous publication [30].

**Table 1 jpm-11-01332-t001:** Detection of HML-2 in cancer and its role as a biomarker or treatment target.

Cancer Type	Biomarker Detected/HML-2 Component Targeted	Observation	Ref
Seminoma cells	Antibodies targeting N terminal of HML-2 gag protein	Protein expressed in cytoplasm of cells and antibodies against this protein were detected in 2–4% patients	[16]
Germ-cell tumors	HML-2 gag and env antibodies in patients	HERV-K/ HTDV gag and env antibodies expressed in germ-cell tumors	[18]
LGL leukemia	Polymorphic HML-2 loci in human genomes	LGL leukemia patients carry higher burden of polymorphic HERV-K proviruses compared to individuals from thousand genomes project of European ancestry	[20]
Melanoma	HML-2 hypomethylation	Independent predictor of disease progression, hypomethylation correlated with reduced disease-free survival	[23]
Colon cancer	HML-2 promoter hypomethylated, higher transcription in tumor, env protein in tumor only	HML-2 hypomethylation in tumor leads to its transcription and translation in tumor tissue	[24]
AT/RT	HML-2 env protein expression in AT/RT patient tissue, HML-2 transcription in AT/RT cells	Targeted decrease of HML-2 expression results in decreased proliferation and viability of ATRT cells	[30]
Ovarian clear cell carcinoma	Hypomethylation of HML-2, transcription of HML-2	Hypomethylation of HERV-K leads to higher expression and was associated with a poor prognosis and platinum resistance	[34]
Hepatocellular carcinoma	HML-2 expression detected in patient samples with qRT-PCR	Level of HML-2 transcription was independent prognostic factor for overall survival rate of hepatocellular carcinoma patients, higher level yielded worse prognosis	[35]
Breast cancer	HML-2 reverse transcriptase protein expression in patient samples	RT protein expression in patients correlates with poor prognosis for disease-free patients and their overall survival	[36]
Rhabdoid tumors	HML-2 Expression in rhabdoid tumors with SMARCB1 mutations	SMARB1 absence leads to re-expression of multiple endogenous retroviruses, including HML-2	[37]
Germ-cell tumors	Retroviral particles bud from cells after treatment with 5-iodo-2’-deoxyuridine and dexamethasone	Germ-cell tumor cells treated with an antiviral pyrimidine analog and the corticosteroid dexamethasone release retroviral particles, but no particles were observed in untreated Tera-1 or Tera-2 cell lines	[40]
Carcinoma in situ	HML-2 transcripts	In situ hybridization identified transcription of HERV-K in carcinoma in situ samples	[41]
Teratocarcinoma cells	Locus-specific expression of HML-2 and transcripts packaged into viral particles	Chr22q11.21 is the locus expressing HML-2 transcripts in Tera-1 cells, and HML-2 RNAs are selectively packaged into HML-2 retroviral particles	[42]
Germ-cell tumors	HML-2 gag/env antibody detection	Concerns 85% of patients with germ. Cell tumors produce antibodies directed against HML-2 env; anti-tumor treatment resulted in a decrease in env antibodies	[43]
Prostate cancer	HML-2 gag antibodies	HERV-K gag antibodies detected in prostate cancer patients	[46]
Breast cancer	HML-2 env protein	Immunotherapeutic potential of anti-human endogenous retrovirus K envelope antibodies	[48]
Ovarian cancer	HML-2 env protein	cytotoxicity of human endogenous retrovirus specific T cells	[49]
Germ-cell tumors	HML-2 antibodies to gag or env	Level of HML-2 ab was stable or elevated in patients with relapse	[56]
Pediatric hepatoblastoma	HML-2 transcription	Upregulated HML-2 transcription in cancer compared to healthy liver	[57]

## Data Availability

Not applicable.

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
