# Peer review of "Endogenous Retroviral Elements in Human Development and Central Nervous System Embryonal Tumors"

_jpm, 2021, doi:10.3390/jpm11121332_

Round 1

Reviewer 1 Report

The authors review the roles of ERV expression in development and cancer. They summarize data regarding the regulation of HML-2 expression and function in pluripotent and stem cells. Then, they discuss HML-2 expression in cancer and its potential use as treatment target and biomarker. This is still an emerging field which deserves to be highlighted. The review is largely well written and cites publications beyond their own work.

Due to the focus of this review on HML-2 (and not all ERVs), the title needs to be changed accordingly.   

p4, lines 16-21:  This should be re-phrased as follow-up study for the mechanism of action. At the moment it reads more like the auhtors would contrast previous findings

p6, lines 31-39: Here the authors discuss their own data, but some conclusion statement would be nice

section 6 ERV expression as prognostic marker:
The authors essentially list many references linking ERV expression with prognosis. Showing this as a table would probably give better overview.

Author Response

Reviewer 1:

Comments and Suggestions for Authors

The authors review the roles of ERV expression in development and cancer. They summarize data regarding the regulation of HML-2 expression and function in pluripotent and stem cells. Then, they discuss HML-2 expression in cancer and its potential use as treatment target and biomarker. This is still an emerging field which deserves to be highlighted. The review is largely well written and cites publications beyond their own work.

Author’s Response: Thank you for your kind review and appreciation of our work. 

Due to the focus of this review on HML-2 (and not all ERVs), the title needs to be changed accordingly.   

Author’s Response: We thank the Reviewer for this suggestion.  While we understand the Reviewer’s desire to specify which endogenous retrovirus our previous work discusses, we do review all the literature relevant to endogenous retroviral elements in embryonal tumors.  Thus, while we focus on HML-2 as an example with the best explained mechanism of involvement, as we state in the manuscript, we believe that endogenous retroviral elements may be related to the development of other embryonal nervous system tumors and that this warrants further investigation. Therefore, we have chosen to leave the title as is. 

p4, lines 16-21:  This should be re-phrased as follow-up study for the mechanism of action. At the moment it reads more like the authors would contrast previous findings

Author’s Response: We thank Reviewer 1 for their careful review and helpful suggestion. We have made the requested change to the manuscript. Please see changes below in red.

In this study, the authors observed a subset of ERV loci were actively expressed and hypothesized that this expression could lead to an anti-tumor response via viral mimicry.39 We were the first to demonstrate the mechanism by which HERV-K HML-2 upregulation occurs in a type of rhabdoid tumor, namely AT/RT.32

p6, lines 31-39: Here the authors discuss their own data, but some conclusion statement would be nice

Author’s Response: We thank Reviewer 1 for their helpful suggestion to emphasize our conclusion in this paragraph. We have added a sentence to enhance emphasis as requested. Please see changes below in red.

Disruption of env expression in AT/RT and prostate cancer cells affects multiple pathways indicating that expression of HML-2 is necessary for their cell viability and proliferation.

section 6 ERV expression as prognostic marker:
The authors essentially list many references linking ERV expression with prognosis. Showing this as a table would probably give better overview.

Author’s Response: We thank Reviewer 1 for the suggestion to organize and compile a table containing the relevant studies in this section.  We have added Table 1 to the end of the manuscript.  

Reviewer 2 Report

The review by Doucet-O'Hare et al is a well-written, clear overview of the contribution of endogenous retroviral elements to the development of embryonal tumors. The authors present a detailed summary of the presence and activation of HERVs in the human genome with a focus on HERV-K HML-2 and AT/RT, and provide arguments for using HML-2 expression as a diagnostic tool and a treatment target.

I have some general points to consider which may be important to the reader of your paper:

- Is it known whether or not HML-2 is expressed from a single locus, or from multiple loci in the tumors described? Can the expressed env RNA be distinguished by sequencing?

- For some rhabdoid tumors, familial occurrence have been described. This could be due to a certain SMARCB1 genotype, or to a polymorphic HML-2 integration. Is anything known about such characteristics?

- There are various pediatric tumors, many outside the brain, with complete loss of SMARCB1 expression. Could you speculate on the contribution of HML-2 expression to malignancy here? And, more specifically, in rhabdoid neoplasms outside the CNS?

- Would the mechanism of SMARC inhibition as described here also be true for SMARCA4-deficient tumors, e.g. has SMARCA4 are similar mode of action to SMARCB1?

- Could SMARCB1 be a general antiretroviral protein? Or a tumor repressor?

Minor comments:

-Section 2: An additional mechanism to repress ERV expression in the early embryo is tRF silencing (for a review, see Cullen & Schorn, Viruses 2020, 12, 792; doi:10.3390/v12080792)

-Page 2, line 17: Homo sapiens, with a capital H. Also, HML-2 probably integrated into the germ line of an ancestor of H. sapiens given that the event took place millions of years ago

-Page 4, line 33: AT/RT

-Sometimes env is written in italics when referring to the gene or to RNA expression (for instance on page 3), sometimes it is not. Please be consistent

Author Response

Reviewer 2:

The review by Doucet-O'Hare et al is a well-written, clear overview of the contribution of endogenous retroviral elements to the development of embryonal tumors. The authors present a detailed summary of the presence and activation of HERVs in the human genome with a focus on HERV-K HML-2 and AT/RT and provide arguments for using HML-2 expression as a diagnostic tool and a treatment target.

Author’s Response: We thank Reviewer 2 for their thoughtful commentary and supportive review of our manuscript. 

I have some general points to consider which may be important to the reader of your paper:

  1. Is it known whether HML-2 is expressed from a single locus, or from multiple loci in the tumors described? Can the expressed env RNA be distinguished by sequencing?

Author’s Response: We thank Reviewer 2 for the suggestion to address the source of HML-2 expression in tumors.  We have added the following paragraph to clarify and expand on this topic.

Identification of the HML-2 loci in the genome expressed in various tumor types is important for understanding the epigenetic implications of their expression as well as identifying which loci may be involved. In AT/RT patient cells, we identified expression from 36 loci from nearly all chromosomes with the exceptions of 14, 15, 18, and 21.32 The most actively transcribed loci were on chromosomes 1 and 19; however, of all expressed loci, only three encoded a full length and possibly functional env protein: transcripts originating from Chr19q11, Chr7pp22.1a, and Chr7p22.1b. 32 When seeking to validate the envelope transcript identified by RNA-sequencing, the reverse transcribed cDNA from a cell line or sample can be cloned into a vector.32 Using Sanger sequencing, we were able to obtain longer reads than with RNA-Seq analysis (700 vs 100 nucleotides) which enabled better alignment of the expressed transcripts to the genome.32 This technique should be employed in additional studies of HML-2 expression in cancer to precisely identify loci which can encode functional HML-2 proteins, such as env,  that contribute to tumorigenesis.

  1. For some rhabdoid tumors, familial occurrence has been described. This could be due to a certain SMARCB1 genotype, or to a polymorphic HML-2 integration. Is anything known about such characteristics?

There are various pediatric tumors, many outside the brain, with complete loss of SMARCB1 expression. Could you speculate on the contribution of HML-2 expression to malignancy here? And, more specifically, in rhabdoid neoplasms outside the CNS?

Would the mechanism of SMARC inhibition as described here also be true for SMARCA4-deficient tumors, e.g., has SMARCA4 are similar mode of action to SMARCB1?

Author’s Response: We thank Reviewer 2 for their insightful thoughts.  We have added the following sentences/paragraphs to expand on all the aforementioned topics.

We have not evaluated the potential contribution of HML-2 transcription from polymorphic elements; however, in Large Granular Lymphocytic (LGL) leukemia, patients have a heavier genomic load of polymorphic HML-2 loci compared to those with a similar genetic background.20 A study like the LGL leukemia patient analysis could reveal the potential importance of polymorphic HML-2 loci in AT/RT and further identify proviruses which produce the env protein. Due to the role of SMARCB1 in the downregulation of HML-2 LTR transcription, any effect on polymorphic HML-2 insertions would not differ from its effect on fixed HML-2 insertions. Therefore, we hypothesize that regardless of the origin of an HML-2 transcript, either from a polymorphic or fixed provirus, SMARCB1 loss would result in higher transcription of any proviruses with C-MYC transcription factor binding sites.

A small percentage of AT/RT cases occur due to the loss of SMARCA4, another core member of the SWI/SNF chromatin remodeling family; moreover, single inherited mutations of either SMARCB1 or SMARCA4 contribute to Rhabdoid Tumor Predisposition Syndrome (RTPS) 1 or 2, respectfully.41 Although no one has evaluated the expression of endogenous retroviral elements in individuals with RTPS1/2, from our work we hypothesize these individuals are likely to have an upregulation of HML-2 transcription due to the absence of functional SWI/SNF regulation at HML-2 LTRs. In addition, we expect HML-2 upregulation will be a feature of non-CNS tumors which lack SMARCB1 expression, especially in tumors with the expression of C-MYC. 

  1. Could SMARCB1 be a general antiretroviral protein? Or a tumor repressor?

Author’s Response: We thank Reviewer 2 for these insightful questions.  SMARCB1 is a tumor repressor and has been shown to inhibit MYC.  We explored how SMARCB1 acts on C-MYC in our original manuscript in scientific reports, “SMARCB1 deletion in atypical teratoid rhabdoid tumors results in human endogenous retrovirus K (HML-2) expression,” this year.  SMARCB1 acts as a general chromatin remodeler for the human genome and plays many roles which include its impact on repression of HML-2 transcription.

Minor comments:

  1. Section 2: An additional mechanism to repress ERV expression in the early embryo is tRF silencing (for a review, see Cullen & Schorn, Viruses 2020, 12, 792; doi:10.3390/v12080792)

Author’s Response: We thank Reviewer 2 for their suggestion, and we have added the appropriate reference to the manuscript.  Please see below.

HML-2 gene regulation is controlled by several mechanisms, including methylation of its promoter (5′ LTR), chromatin remodeling, binding by transcription factors, and repression via transfer RNA derived fragments (tRFs).21−25

  1. Page 2, line 17: Homo sapiens, with a capital H. Also, HML-2 probably integrated into the germ line of an ancestor of H. sapiens given that the event took place millions of years ago

Author’s Response: We thank Reviewer 2 for their astute observation, and we have made an adjustment to the sentence.  Please see below.

Human Endogenous Retrovirus K (HERV-K), subtype HML-2, incorporated into the germline of ancestors of Homo sapiens millions of years ago and have been maintained in our genome.15

  1. Page 4, line 33: AT/RT

Author’s Response: We thank Reviewer 2 for their careful reading of the manuscript.  We have added in the / as suggested, please see below.

These findings coupled with others suggest a possible role for HML-2 in tumors which possess an immature undifferentiated morphology like AT/RT.

  1. Sometimes env is written in italics when referring to the gene or to RNA expression (for instance on page 3), sometimes it is not. Please be consistent

Author’s Response: We thank Reviewer 2 for their attentive review, we have now changed all instances of env in the manuscript so that they are uniform.